# Formation of Majorana fermions in finite-size graphene strips

**Vardan Kaladzhyan[1,2]⋆ and Cristina Bena[1,2]**

**1** Institut de Physique Théorique, CEA/Saclay, Orme des Merisiers,
91190 Gif-sur-Yvette Cedex, France
**2** Laboratoire de Physique des Solides, CNRS, Univ. Paris-Sud, Université Paris-Saclay,
91405 Orsay Cedex, France

⋆ vardan.kaladzhyan@cea.fr

## Abstract

We investigate the formation of Majorana fermions in finite-size graphene strips with open boundary conditions in both directions, in the presence of an in-plane magnetic field and in the proximity of a superconducting substrate. We show that for long enough strips the Majorana states can form in the presence of a Rashba-like spin-orbit coupling, as well as in the presence of an inhomogeneous magnetic field. We find that, unlike infinite graphene ribbons in which Majorana states arise solely close to the bottom of the band and the Van Hove singularities, for finite-size systems this can happen also at much smaller doping values, close to the Dirac points, and depends strongly on the type of the short edges of the systems (e.g. armchair vs. zigzag), as well as on the width of the ribbons.



## 1 Introduction

The formation of Majorana fermions has been proposed and searched for in various one-dimensional and two-dimensional systems [1–28]; most proposals concern systems with strong spin-orbit coupling in the presence of a Zeeman magnetic field, such as semiconducting InAs and InSb wires, brought in the proximity of a superconducting substrate [29–31]. Alternatively it has been shown that the spin-orbit coupling can be replaced by an inhomogeneous magnetic field, such as the one generated by a network of magnetic impurities deposited on a substrate. Experiments to test the formation of Majorana states in magnetic-impurity networks have been performed and seem to be consistent with the formation of zero-energy end-states [32]. However the realization of such systems remains difficult and finding alternative methods is an important task in the hope to realize systems in which the Majorana states can be obtained easier and in a reproducible manner, and in which they can be straightforwardly manipulated.

Graphene has the enormous advantage of being easy to fabricate and manipulate. [33, 34] However, its almost negligible spin-orbit, [35–38] as well as the difficulty to obtain large dopings[1], do not make graphene a choice Majorana candidate. In previous works, it has been proposed that Majorana states can form in infinite graphene ribbons either in the presence of a spin-orbit coupling, or of an inhomogeneous magnetic field [39, 40], however these proposals present an important drawback in the necessity to access dopings close to the bottom of the band or the Van Hove singularities [39–41], both being far from experimental reach. Moreover, an infinitely-long ribbon is not a realistic description of an experimental system, and the finite longitudinal dimension needs to be taken into account in the formation of the Majorana fermions, together with the transversal one.

To overcome these problems, in the present work we focus on finite-size graphene strips (see Fig. 1). Such strips can be fabricated quite easily [42–45]. In a recent proposal it has been shown using a single subband approximation that finite-size zigzag graphene strips may support Majorana fermions for not too large doping values [46]. This analysis is valid for relatively thin strips. In what follows we study both zigzag and armchair graphene strips in both rotating and uniform in-plane magnetic fields using a general tight-binding exact diagonalization approach that allows us to take into account not only the effects of the lowest subband, but of the entire spectrum, as well as the influence of the ribbon width on the formation of Majoranas. We thus obtain the topological phase diagrams for graphene strips over large ranges of parameters.

The first system we consider is a graphene strip with a non-zero Rashba spin-orbit coupling, under a uniform in-plane Zeeman magnetic field and in the proximity of a superconducting substrate. Considering an in-plane field has the advantage of not giving rise to significant orbital effects. We use numerical tight-binding methods [47] and the Majorana polarization tool [48–50] to calculate the topological phase diagram, i.e. the number of Majorana states as a function of the chemical potential and the applied magnetic field. We find that the most efficient configuration to form an odd number of Majorana pairs (i.e a topologically non-trivial

---

[1]The average doping in graphene is close to the center of the band and the Dirac point, parameter region which does not seem to facilitate the formation of Majoranas.

state) at low dopings is that of a zigzag strip (zigzag short edges and armchair long edges), with a magnetic field parallel with the long axis. For this case we show that the required magnetic field and chemical potential may be achievable experimentally. While the required values of the spin-orbit are still unrealistic for a clean isolated monolayer graphene, it may be possible to achieve sufficiently large spin-orbit values for curved graphene sheets [51, 52] or nanotubes [12, 53, 54] or in the presence of metallic substrates [55].

We then consider a graphene strip subject to an inhomogeneous (e.g. rotating) in-plane magnetic field. This configuration does not require a non-zero spin-orbit coupling in order to give rise to Majorana states and thus it may be a more straightforward alternative. We find multiple combinations of rotating wave-vector values and edge types that can give rise to Majorana states under realistically achievable values of the magnetic field. The magnetic field needs to rotate along the long-axis of the strip, and can be uniform along the short axis. Such a configuration can be achieved for example by the deposition of a series of magnetic fingers on top of graphene [56], or by intercalating a lattice of magnetic atoms between graphene and the substrate [57].

The paper is organized as follows: in section 2 we present the tight-binding model describing graphene in the presence of magnetic field, spin-orbit coupling and SC proximity, as well as the type of configurations we consider. In section 3 we describe the technique we use to test the formation of Majorana states, i.e. a numerical diagonalization of the tight-binding Hailtonian and the calculation of the Majorana polarization (MP). In section 4 we present our results for the formations of Majorana states in different configurations. We discuss the experimental relevance of our results in section 5, and we conclude in section 6.

## 2 Model

We consider a hexagonal-lattice tight-binding Hamiltonian, with a spin-orbit Rashba coupling $\alpha$, subject to a magnetic field $\vec{B}(\vec{r})$ and brought in the proximity of a superconducting substrate which is assumed to induce an on-site SC pairing. We write down the corresponding Bogoliubov-de-Gennes Hamiltonian in the Nambu basis, $\Psi_j = (\psi_{j,\uparrow}, \psi_{j,\downarrow}, \psi_{j,\downarrow}^\dagger, -\psi_{j,\uparrow}^\dagger)^T$ where $\psi_{j,\sigma}^\dagger$ creates a particle of spin $\sigma$ at site $j$. We use the Pauli matrices $\vec{\sigma}$ for the spin subspace and $\vec{\tau}$ for the particle-hole subspace. The full Hamiltonian is

$$H = H_0 + H_B + H_R. \tag{1}$$

The first term of the Hamiltonian is given by

$$H_0 = \sum_j \Psi_j^\dagger \left[ -\mu\tau^z - \Delta\tau^x \right] \tilde{\Psi}_j - t \sum_{\langle i,j \rangle} \Psi_i^\dagger \tau^z \Psi_j, \tag{2}$$

where $\mu$ is the chemical potential, $t$ the hopping strength, and $\Delta$ the induced superconducting pairing. We set $t = \hbar = 1$ throughout. We also take the distance between nearest neighbours $a = 1$ and we denote the distance between the nearest atoms of the same sublattice $d = \sqrt{3}$ (see Fig. 1).

The second term in Eq. (1) describes the effect of the Zeeman magnetic field of strength $B$ and is given by

$$H_B = \sum_j \Psi_j^\dagger \vec{B}_j \cdot \vec{\sigma} \Psi_j. \tag{3}$$

The term corresponding to the effective Rashba spin-orbit interaction of strength $\alpha$ can be written as:

$$H_R = i\alpha \sum_{\langle i,j \rangle} \Psi_i^\dagger \left( \vec{\delta}_{ij} \times \vec{\sigma} \right) \cdot \hat{z} \tau^z \Psi_j. \tag{4}$$

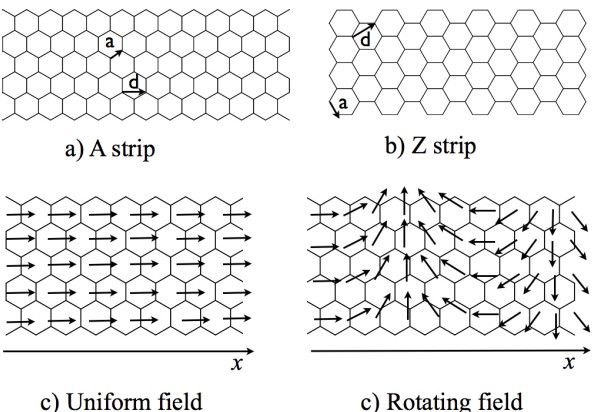

Figure 1: a) Graphene strip with short armchair edges (A strip) b) Graphene strip with short zigzag edges (Z strip) c) Magnetic field configuration for an in-plane uniform field d) Magnetic field configuration for a rotating in-plane field.

We define the particle-hole operator $\mathscr{C} = \mathrm{e}^{\mathrm{i}\zeta} \, \sigma^y \tau^y \hat{K}$, where $\hat{K}$ is the complex-conjugation operator, and $\zeta$ is an arbitrary phase. The Hamiltonian anticommutes with this operator, $\{\mathscr{C}, H\} = 0$, and $\mathscr{C}^2 = 1$.

To understand the energy scales involved we note here that for $B = \Delta = \mu = 0$ the square lattice has a bandwidth of $8t$, while the hexagonal lattice has a bandwidth of $6t$ and a Van Hove singularity at $t$ (experimentally the bandwidths are of the order of a few $eV$).

We consider first a uniform magnetic field $\vec{B}(\vec{r}) = \vec{B}$, and in particular we focus on an in-plane magnetic field parallel with the longer dimension of the strip $\vec{B} = B\hat{x}$. Alternatively we consider an in-plane rotating magnetic field $\vec{B}(\vec{r}) = B[\hat{x} \cos(\vec{q} \cdot \vec{r}) + \hat{y} \sin(\vec{q} \cdot \vec{r})]$, and a rotation direction parallel to the long axis of the strip, $\vec{q} \parallel \vec{x}$. The physical origin of the magnetic field may be for example an applied Zeeman field via a network of magnetic fingers deposited on top of graphene [56], or a network of magnetic adatoms intercalated between the strip and the substrate [57]. The configurations that we consider are depicted in Fig. 1, i.e. a) strips with armchair edges on the short side, and zigzag edges on the long side ($\hat{x}$) (we will denote them as A strips), b) strips with zigzag edges on the short side and armchair edges on the long side (Z strips). The magnetic field configurations that we focus on are depicted in Fig. 1c (uniform field) and Fig. 1d (rotating field).

## 3 Method

The tool that we use to test the formation of Majorana states in these systems is a numerical analysis of the eigenstates of the tight-binding Hamiltonian (performed with the help of the numerical code MatQ [47]). We write the corresponding wavefunctions as $\psi_{\vec{r}}^T \colon \{u_{\vec{r}\uparrow}, u_{\vec{r}\downarrow}, v_{\vec{r}\downarrow}, v_{\vec{r}\uparrow}\}$. We use the Majorana-polarization criterion introduced in Ref. [28, 48–50], applied to the low-energy eigenstates of our systems. The way in which the MP criterion works is as follows: a Majorana state is an eigenstate of the particle-hole operator, therefore a Majorana state localized inside a spatial region $\mathscr{R}$ must satisfy $C = 1$ where

$$C = \frac{\left| \sum_{j \in \mathscr{R}} \langle \Psi | \mathscr{C}_j | \Psi \rangle \right|}{\sum_{j \in \mathscr{R}} \langle \Psi | \hat{r}_j | \Psi \rangle} \tag{5}$$

is the appropriately normalized magnitude of the integral of the Majorana polarization vector $P_j(\vec{r})$ over the spatial region $\mathscr{R}$, with

$$P_j(\vec{r}) = -2 \sum_\sigma \sigma u_{\vec{r}\sigma} v_{\vec{r}\sigma}. \tag{6}$$

In general we take $\mathscr{R}$ to correspond to half the graphene strip, divided along the shorter length. By numerically solving the tight-binding Hamiltonian in Eq. (1) and plotting $C$ as a function of the system parameters we can accurately recover the appropriate topological phase diagram. The topological phases are characterized by $C = 1$, the non-topological ones by $C \approx 0$.[2]

An interesting situation arises when more than one pair of Majorana states is present. In our calculations we sum the MP over the states with the lowest energies, thus counting the number of Majorana modes present. As described also in previous references, the states with an even number of Majoranas are not topologically protected and can easily be destroyed by disorder, while the states with an odd number of Majoranas are topologically protected.

## 4 Results

### 4.1 Graphene strips with spin-orbit coupling subject to a uniform in-plane magnetic field

We begin by presenting the results for graphene strips with spin-orbit coupling in the presence of a uniform magnetic field. We focus on the case of an in-plane magnetic field parallel to the longer axis of the strip (see Fig.1c). A similar analysis for the formation of Majorana fermions in square-lattice strips has been performed in Ref. [50] for a perpendicular field and in Ref. [28] for an in-plane field. In the previous analyses it was found that for a purely one-dimensional system the Majorana states were forming mainly for $B \gtrsim \sqrt{(\mu-2)^2 + \Delta^2}$, while for a two-dimensional system (infinite ribbon) for $B \gtrsim \sqrt{(\mu-4)^2 + \Delta^2}$. For finite-size strips the corresponding topological phases were described in Refs. [50] and [28], and were composed of multiple discrete topological regions in the $(\mu, B)$ parameter space, the details depending strongly on the direction of the magnetic field (in-plane or perpendicular to the lattice).

### 4.1.1 Infinite ribbons

For a hexagonal lattice the case of an infinite ribbon subject to a perpendicular uniform magnetic field was analyzed in Ref. [39]. In Fig. 2 we present the topological phase diagram in a ribbon with zigzag edges (a similar situation is recovered for armchair edges), as well as the band structure for a set of parameters corresponding to the formation of Majoranas. The topological phase diagram was obtained by using the MatQ code [47] to diagonalize the Hamiltonian of the nanoribbon, and subsequently calculating the corresponding Majorana polarization of the lowest energy states [28, 48–50]. The pink regions correspond to lowest-energy edge states with a MP of 1, while the white region to a zero MP. We find that the condition to have Majorana states for an in-plane field remains qualitatively the same as for a perpendicular field: $B \gtrsim \sqrt{(\mu-1)^2 + \Delta^2}$ and $B \gtrsim \sqrt{(\mu-3)^2 + \Delta^2}$, thus for small magnetic fields Majorana states tend to form for chemical potentials close to the Van Hove singularity and to the bottom of the band (these values are actually slightly shifted and do not correpond exactly to $\mu = 1$ and $\mu = 3$, as it can be seen in Fig. 2). The exact boundaries of the topological phases [40]

---

[2] In general one can have many different situations in which some states can acquire a finite $C < 1$, even at finite energy, but only the states with $C \approx 1$ are Majorana. Thus in many of our calculations we will only take into account the states with a value of $C$ larger than a specific cutoff, for example $C = 0.8$.

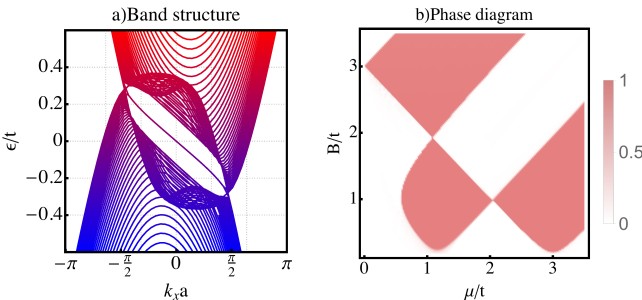

Figure 2: a) The band structure for a zigzag ribbon of $W = 86a$, with in-plane field for $\mu = 2.9t$, $B_y = 0.5t$, $\alpha = 0.5t$, $\Delta = 0.2t$. Note the formation of chiral Majorana states and the overall gapless character of the band structure. b) Topological phase diagram for a zigzag ribbon of $W = 260a$ as a function of $\mu$ and $B$ for $\alpha = 0.5t$, and $\Delta = 0.5t$. We plot the MP of the lowest energy state, thus a value of 1 corresponds to a topological phase (pink), while a value of 0 to the trivial one (white).

depend on the value of the spin-orbit coupling. A detailed analysis of the dependence of the topological phase diagram on the value of the spin-orbit coupling for a perpendicular magnetic field was presented in Ref. [39]. We would like to emphasize that unlike the case of a perpendicular field [39], the band structure is gapless (same as for a square lattice with an in-plane field [28]), and chiral Majorana edge modes can form, as shown in Fig. 2a.

The main point to note is that in order to obtain Majorana states it is required to dope the graphene sheet up to the Van Hove singularity or to the top or bottom of the band, which is not feasible experimentally, or alternatively achieving magnetic fields of the order of the bandwidth, which is unrealistic.

### 4.1.2 Graphene strips

The required dopings to form Majorana states can be reduced by considering graphene strips of finite-size. In Fig. 3 we present the topological phase diagrams for strips with different types of edges for a very large value of the spin-orbit $\alpha = 0.5t$. The phase diagrams are determined by adding up the MP vectors over half of the strips and summing over all the lowest energy modes that have a MP larger than a given cutoff, here taken to be 0.8. Note that our plots show the number of Majorana pairs, not the topological character of the system (the states with even numbers of Majoranas are topologically trivial and can be destroyed easily by adding a small amount of disorder [28, 40]). We note that the region in the parameter space which allows the formation of Majoranas is greatly extended compared to the infinite ribbon case. Thus, particularly for an A-type strip (left column) we see that we can form Majorana states even at chemical potential close to zero (undoped graphene), and for not too large magnetic fields.

### 4.1.3 Spin-orbit effects

In Fig. 4 we also show the topological phase diagram for a smaller value of the spin-orbit $\alpha = 0.1t$. To avoid the use of very large systems (cumbersome to diagonalize numerically) required to eliminate the numerical errors at large magnetic fields, in Fig. 4 we set the maximum value for the magnetic field to $B = t$.

Note that for smaller spin-orbit values, the Z strips are actually more convenient than the A strips to form an odd number of Majorana pairs. Thus a single-Majorana-mode phase can be achieved for not too large values of the chemical potential of the order of $0.2 - 0.3t$. This is consistent also with the observation of the single-band analysis of Ref. [46]. This value depends strongly on the width of the strip, this will be discussed in section 4.1.4. While the

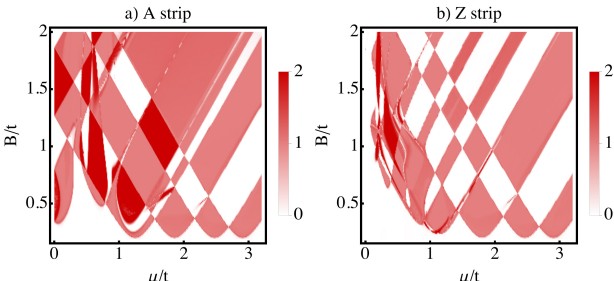

Figure 3: The total MP as a function of the chemical potential and magnetic field for A strips (left panel) and Z strips (right panel). We consider a ribbon of width of $W = 5a$ and length $L = 520a$, for $\alpha = 0.5t$ and $\Delta = 0.2t$.

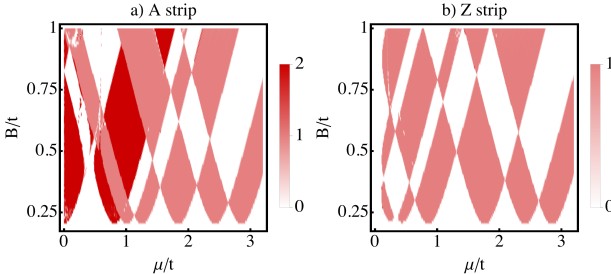

Figure 4: The total MP as a function of the chemical potential and magnetic field for A strips (left panel) and Z strips (right panel). We take $W = 5a$, $L = 520a$, $\alpha = 0.1t$ and $\Delta = 0.2t$.

values for the SC gap and the spin-orbit coupling used in this section remain unrealistically large in order to limit the system size and the computation time, we discuss the possibility to obtain Majorana states with experimentally realistic parameters in section 5.

We note also that the value of the spin-orbit coupling has a strong influence on the phase diagram, as expected (see Ref. [40]), and it is even more advantageous to have a not-too-large spin-orbit coupling, especially for the Z strips. In Fig. 5 we plot the MP as a function of the spin-orbit coupling value and the magnetic field for a small chemical potential, for both A strips ($\mu = 0.1t$) and Z strips ($\mu = 0.2t$). We note that for the A strips phases with an even number of Majorana states dominate, except for very large spin-orbit couplings, however for the Z strip the low-spin-orbit regions are susceptible to form rather a single pair of Majorana states.

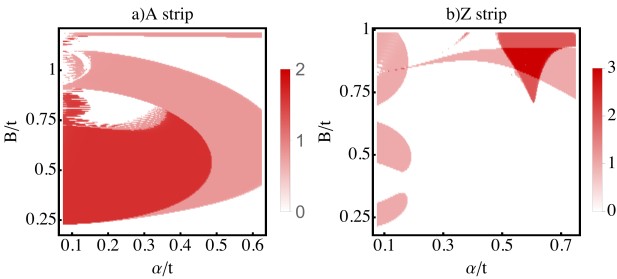

Figure 5: The total MP as a function of the spin-orbit value and magnetic field for A strips (left panel, $\mu = 0.1t$), and for Z strips (right panel, $\mu = 0.2t$). We take $W = 5a$, $L = 260a$, and $\Delta = 0.2t$.

### 4.1.4 Dependence on ribbon width

We also consider the influence of the system width on the phase diagram. In Fig. 6 we plot the topological phase diagram for a wider system $W = 15a$. We note that the formation for Majorana bound states(MBS) can also be achieved for small values of $\mu$ and $B$. We expect thus that our results remain valid for even wider ribbons, as long as the long edge stays much larger than the inverse spin-orbit length [28].

To test this hypothesis we have performed an analysis of the minimal value for which the Majorana states can form at $V_Z \approx \Delta$ as a function of the width of the ribbon. In Fig. 7a) we plot this value as a function of inverse-ribbon-width $\pi/2W$ (red circles and black squares). The dashed red line correspond to the gap of the parent infinite non-metallic zigzag ribbon of the same width $W$, i.e $E_{\text{gap-semiconductor}} = \pi/2W$ [58], see Fig. 7b), and the dashed black line to the bottom of the second band for a metallic zigzag ribbon ($E_{\text{second-band-metal}} = 3\pi/2W$) [58] see Fig. 7c). Indeed we see that the chemical potential minima obtained in the numerical simulations correspond well to fillings close to the bottom of the first band in the case of a semiconducting ribbon, and of the second band (the first one being gapless) for the case of the metallic ones. Also we note the alternance of two semiconducting ribbons and one metallic one when increasing the width by one unit cell as expected [58].

We should note that when the width of the ribbon increases, such that the width of the ribbon is becoming comparable to the spin-orbit length (equivalently this can be achieved by increasing the value of the spin-orbit coupling) the value of the magnetic field required to form Majorana states form at $\mu = \pi/2W$ is actually increasing quite substantially and is no longer equal to $\Delta$ (see for example Fig. 3b) in which $\alpha = 0.5$). Thus for larger spin-orbit couplings, or equivalently, for wider ribbons, MBS can only form at larger magnetic fields or larger chemical potentials. This was also observed in Fig. 5b where the formation of MBS is studied as a function of the value of the spin-orbit value. The empty circles in Fig. 7a) denote the minimal chemical potentials at which MBS form, but they do not account for the formation of MBS at $V_Z \approx \Delta$ (as it is the case for the filled circles) but for the formation of MBS at $V_Z > \Delta$.

The blue triangles correspond to the second minimal value of $\mu$ giving rise to MBS at $V_Z \approx \Delta$, and the blue dashed line to the bottom of the second band of a zigzag ribbon ($E_{\text{second-band-semiconductor}} = \pi/W$). The agreement is very good, confirming that the MBS always form close to the quadratic points of a given band structure, as also observed in Ref. [28].

Thus we conclude that MBS form close to the bottom of the first band of the parent infinite semiconducting nanoribbons, and close to the bottom of the second band for the metallic ones (the first one being gapless). Higher order MBS form close to the minima of the higher bands. The minimal chemical potential thus decreases as the inverse of the ribbon width. However, when the width of the ribbon becomes comparable to the spin-orbit length, the required value of the magnetic Zeeman field required to form MBS in the lowest bands increase, thus becoming eventually more efficiently to form MBS at low $V_Z$ in the higher bands at larger chemical potentials.

### 4.1.5 Perpendicular magnetic field

We also study the effects of the direction of the magnetic field for the formation of MBS. As described in [28] there can be large differences between in-plane fields and fields perpendicular to the plane of the ribbon. In Fig. 8 we give the topological phase diagrams for a Z strip for the same parameters as those in Fig. 3b) for an in-plane field. We see that indeed there are significant differences, in particular we see that the region between $\mu = 1$ and $\mu = 3$ the required magnetic field to form MBS is larger for a perpendicular field than for an in-plane field.

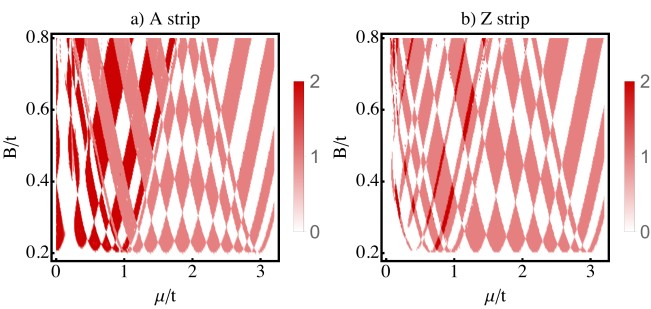

Figure 6: The total MP as a function of the chemical potential and magnetic field for A strips (left panel) and Z strips (right panel). We take $W = 15a$, $L = 432a$, $\alpha = 0.1t$ and $\Delta = 0.2t$

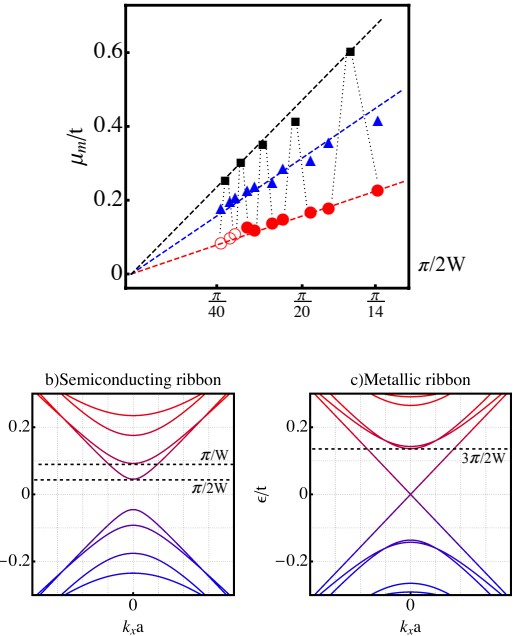

Figure 7: a) The dependence of the chemical potential required to form MBS as a function of ribbon width. The red circles correspond to semiconducting ribbons, while the black squares to metallic ones. The blue triangles correspond to the second lowest value of the required chemical potential. For all these points we consider the formation of MBS at $V_Z \approx \Delta$. The empty red circles correspond to the formation of MBS at $V_Z \approx \Delta$. The dashed red line corresponds to the gap of a semiconducting ribbon $E_{\text{gap-semiconductor}} = \pi/2W$ , the dashed black line to the minimum of the second band of a metallic ribbon $E_{\text{second-band-metal}} = 3\pi/2W$, while the blue dashed line to the minimum of the second band for a semiconducting ribbon $E_{\text{second-band-semiconductor}} = \pi/W$. b) The band structure for a parent infinite semiconducting strip of width $W$. c) The band structure for a parent infinite metallic strip of width $W$.

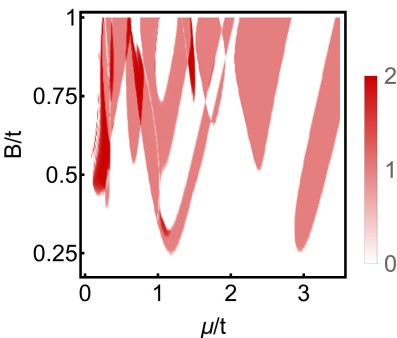

Figure 8: The total MP as a function of the chemical potential and a magnetic field perpendicular to the plane for Z strips. We consider a ribbon of width of $W = 5a$, for $\alpha = 0.5t$ and $\Delta = 0.2t$, same as in Fig. 3b) for an in-plane field.

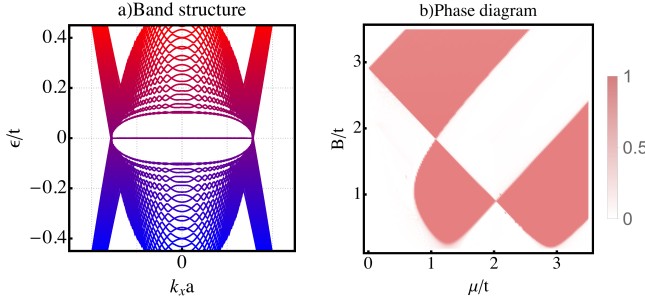

Figure 9: a) Band structure for a graphene nanoribbon with a magnetic field rotating in the plane of the ribbon, with an wave vector $2\pi/5d$ parallel to the edges of the ribbon, for $\mu = 3t$, $B = t$ and $\Delta = 0.2t$, $W = 86a$. b) Phase diagram as a function of $\mu$ and $B$ for $\Delta = 0.2t$, $W = 173a$.

## 4.2 Graphene strips in the presence of a rotating in-plane magnetic field

As shown in previous works, Majorana states can also form in the presence of inhomogeneous magnetic fields. For graphene ribbons this has been studied in Refs. [40] and [46]. It was shown [40] that typically flat Majorana bands form in such systems, and that the Majorana states at low fields tend to form close to the values of the chemical potential of $\mu \approx t$ and $\mu \approx 3t$, same as for the case of a uniform field and spin-orbit coupling. In Fig. 4 we present the phase diagram for a configuration in which the magnetic field rotates in the plane of the lattice, with a wavevector parallel to the edge of the ribbon, and a typical topological band structure showing Majorana zero-energy flat bands.

We can see that, same as for spin-orbit case, the topological phases arise for unrealistic values for the parameters. However we expect that using a finite-size strip will help overcome this problem by extending the topological regime with respect to the infinite case. Moreover, we expect that in the finite-size strips the formation of a large number of Majorana modes is possible, as a reminiscent of the flat bands in the infinite case.

Indeed, in Fig. 10 we plot the topological phase diagram for an A strip as a function of the chemical potential and a magnetic field of amplitude $B$ rotating in the $xy$ plane, and whose rotation axis is parallel with the long edge of the strip (see Fig. 1d). We again sum the Majorana polarizations of the lowest-energy states (same as before we consider only the states that have a MB larger than 0.8). We take $\vec{q} = (2\pi/3d, 0)$ (a,c) and $\vec{q} = (2\pi/10d, 0)$ (b,d), (as described in section 2, $d = \sqrt{3}a$ is the distance between two nearest atoms belonging to the same sublattice). We focus both on thinner and wider strips ($W = 5a$ and $W = 15a$). We find

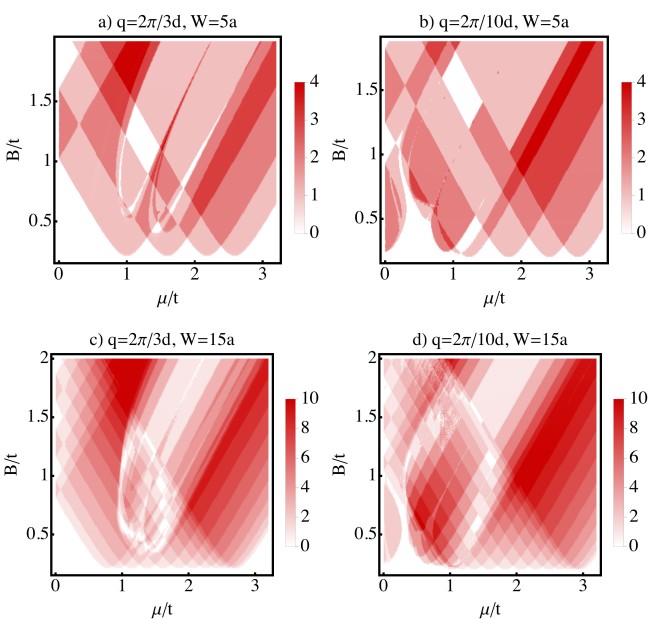

Figure 10: A strips: Topological phase diagrams for $W = 5a$ (top) and $W = 15a$ (bottom). We consider a wavevector of $q = 2\pi/3d$ (left column, $L = 86a$), and $q = 2\pi/10d$ (right column, b) $L = 346a$ and d) $L = 173a$ ). We take $\Delta = 0.2t$.

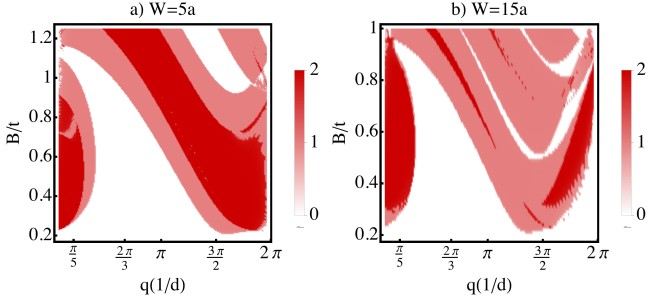

Figure 11: a) Dependence of total MP with the amplitude of the magnetic field $B$ (in units of $t$) and $q$ (in units of $2\pi/d$) for A strips. We use a) $W = 5a$ and b) $W = 15a$ strips, with $L = 173a$ and $\Delta = 0.2t$.

that for large rotation wavevectors such as $\vec{q} \approx (2\pi/3d, 0)$, we have extended regions in which at least one Majorana pair forms for not too large chemical potentials. However, same as for the spin-orbit case, for this particular choice of parameters a significant part of this region corresponds to an even number of Majoranas and thus to states that are not topologically protected.

To identify the easiest way to achieve topological odd-parity states, in Fig. 11 a) we plot the dependence of the MP as a function of $B$ and $q$ for a small chemical potential $\mu = 0.1t$. Indeed, we see that for a very-rapidly varying magnetic field $q \approx 3\pi/2$ (corresponding to a periodicity of the magnetic field of approximatively $1.25d$ one can form odd-parity topological Majorana phases at very small chemical potentials. This limit is explored in more details in section 5 for a physical value of the superconducting gap, $\Delta \approx 1 meV \approx 0.001t$.

For the Z strips we redo a similar analysis for $\vec{q} = (2\pi/3d, 0)$ and $\vec{q} = (2\pi/10d, 0)$, and for $W = 5a$ and $W = 15a$. Thus in Fig. 12 we plot the phase diagram as a function of $B$ and $\mu$, while in Fig. 13 we plot the dependence of the number of Majorana pairs as a function of $B$ and $q$ for a value of the chemical potential of $\mu = 0.5t$. We note that the formation of topological

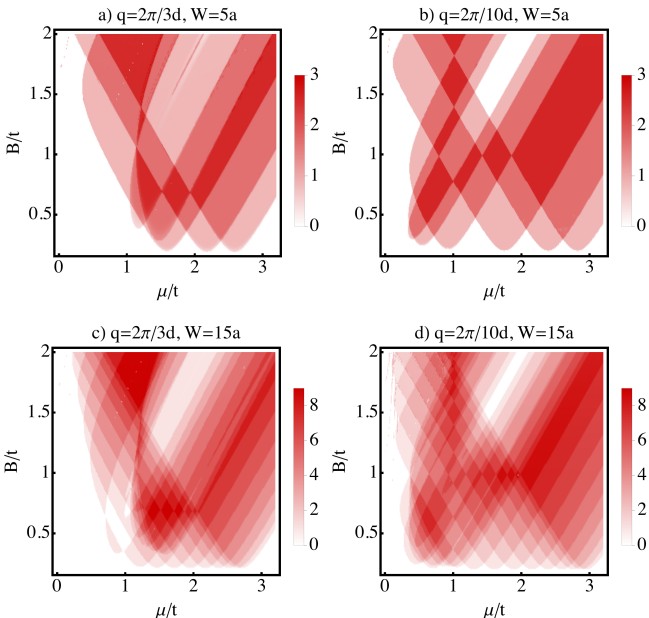

Figure 12: Z strips: Topological phase diagrams for strips with $W = 5a$ (upper row) and $W = 15a$ (lower row). We consider a wavevector of $q = 2\pi/3d$ (left column, $L = 86a$), and $q = 2\pi/10d$ (right column, b)$L = 346a$ and d)$L = 173a$). As before we take $\Delta = 0.2t$.

states requires in this situation slightly larger chemical potentials of the order of $0.3 - 0.5t$.

## 5 Comparison with realistic situations

We will focus on two situations, that we believe are easiest to achieve experimentally. The first is that of a Z strip, for which we take $W = 5a$. We consider a SC gap $\Delta \approx 1meV \approx 0.001t$, and a spin-orbit $\alpha = 0.05t \approx 50meV$. Such values for spin-orbit couplings have been predicted to arise for example in graphene on specific metallic substrates [55]. We plot the MP as a function of the magnetic field and chemical potential and we find a topological region centered around $\mu = 0.23 - 0.24t$, for $B > 0.001t \approx 10T$. Note that the magnetic field value can be reduced if one can increase the value of the $g$ factor, such as e.g. in InAs wires. At present in-plane fields of a few Tesla can be achieved experimentally without destroying the superconductivity

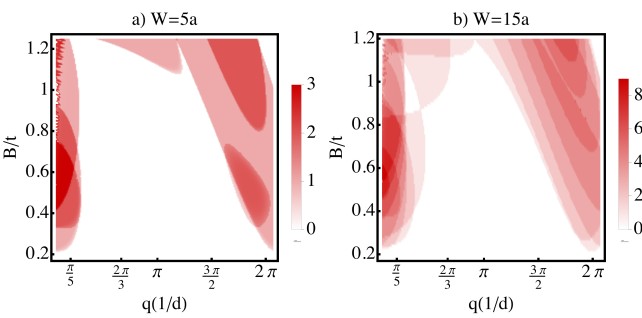

Figure 13: a) Dependence of total MP with the amplitude of the magnetic field $B$ (in units of $t$) and $q$ (in units of $2\pi/d$) for Z strips. Use a) $W = 5a$ and b) $W = 15a$ strips, with $L = 173a$, and $\Delta = 0.2t$

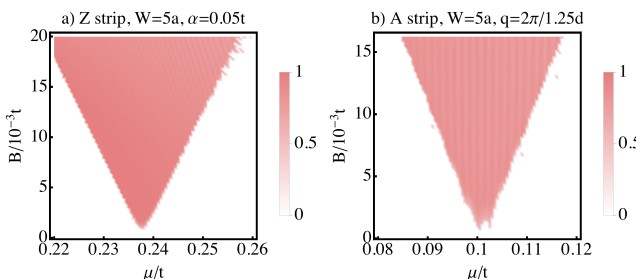

Figure 14: Dependence of the MP with $B$ and $\mu$ for a strip with $W = 5a$, $L = 1557a$, $\Delta = 0.001t$ for a) a Z strip with $\alpha = 0.05t$. Note the formation of Majoranas for a region centered around $\mu = 0.23 - 0.24t$, for $B > 0.001t$ b) an A strip with $q = 5 \approx 2\pi/(1.25d)$. Note the formation of Majoranas for a region centered around $\mu = 0.1t$, for $B > 0.001t$.

[59]; the values necessary in our simulation are of the same order. A similar topological region, not shown here, is found around $\mu = 0.42t$. Of course, this is still a very rough approximation, fully realistic simulations taking into account more complex factors such as higher-order hopping in graphene, wider and longer ribbons, and more complex phenomena such as edge reconstruction, need to be performed. For the parameters considered here our results are consistent with the single-band results of Ref. [46]; it would be especially interesting to use our approach to study also wider ribbons for which the single-band model may not be sufficient to capture all the physics.

For the rotating field situation we take $q = 5 \approx 2\pi/(1.25d)$ and $\Delta = 0.001t \approx 1meV$. We consider an A strip with $W = 5a$ and $L = 1557a$, and we plot the dependence of MP as a function of $B$ and $\mu$. Again we note a topological region for $B > 0.001t \approx 10T$. Same as before, these are only indicative results showing that experimentally achievable values may be attained, however fully realistic calculations are necessary in order to predict the exact regions in the parameter space for which the formation of odd-parity Majorana states is possible; this is however beyond the scope of the present work.

## 6 Conclusions

We study the formation of Majorana states and of topologically non-trivial phases in finite-size graphene strips proximitized by a superconductor, in the presence of either a uniform in-plane magnetic field and of Rashba spin-orbit coupling, or of a rotating in-plane magnetic field. Such conditions can be achieved in either curved graphene strips or tubes, in the presence of a metallic substrate or of magnetic adatoms; lately these possibilities have received an increasing experimental attention. We use a tight-binding model which allows us to go beyond the single-band approximation and to take into account the full spectrum of the system. The main advantage of using finite-size graphene strips is that that they are easy to fabricate, as well as to proximitize, moreover the in-plane magnetic fields do not have any destructive orbital effects in the formation of Majoranas. Also, the required doping of the graphene strips are much smaller than for infinite ribbons. We have shown that Majorana modes can form in various setups, for strips with different types of edges, zigzag and armchair, for different magnetic field configurations, and for experimentally accessible parameters (superconducting gap, doping, spin-orbit coupling and magnetic field). We have also studied the dependence of the formation of MBS on the width of the strips.

# Acknowledgements

We would like to thank Nicholas Sedlmayr, Mark-Oliver Goerbig, Jelena Klinovaja and Daniel Loss for useful comments and fruitful discussions.

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
