# Peer review of "Formation of Majorana fermions in finite-size graphene strips"

_SciPost Physics, doi:SciPost Phys. 3, 002 (2017)_

## Round 2 · Referee Report · Anonymous (Referee 1) · 2017-4-7

Strengths

Accessible and well readable. Timely and interesting topic. Detailed numerical study. Good illustration with figures.

Weaknesses

1.) Marginal originality. The setup at hand has been investigated in various previous publications. In particular Ref. [47] investigates almost the same systems just with partly different methods. The new physical insights of this specific manuscript are very limited.

2.) Large parts consist of blackbox numerical analysis without clear physical pictures. For example, the physics behind observations such as the importance of the ribbon width for the stability of the Majoranas is hardly discussed.

3.) No immediate experimental relevance. As a main motivation for the the present study, the authors state the need for finding experimentally more accessible/realistic material systems hosting Majorana bound states. However, as reflected in the conclusion "As a future extension of the present work we propose to perform fully-realistic studies...", the manuscript falls short of really answering relevant experimental questions such as localization and coherence properties of the Majoranas, realistic numbers for the involved energy gaps, etc.

Report

The authors study the occurrence of Majorana bound states in graphene ribbons in proximity to a conventional superconductor. They consider two situations, one which relies on strong spin orbit coupling (possibly realized for graphene on a metallic substrate) and one where a spatially rotating magnetic field generates similar effects (proposed in Ref. [47]). While I generally find the topic very interesting and the present analysis seems sound in content, its low originality and limited experimental relevance in my opinion makes it a candidate (upon minor revision) only for publication in the Tier III section of this journal. For my specific comments and amendments, see the other fields of this review form.

Requested changes

I ask the authors to clearly state how the phase diagram in Fig. 2b) is computed. Furthermore, a clear physical discussion of how the finite size of the ribbons affects their topological properties would be in order, given that this point is strongly emphasized in the introduction.

---

## Round 3 · Author Response

Dear Editor,

We would like to thank the referee for the report, certain comments have helped us to improve the manuscript. Nevertheless, we would like to point out that we do not agree with some of the statements.

“1.) Marginal originality. The setup at hand has been investigated in various previous publications. In particular Ref. [47] investigates almost the same systems just with partly different methods. The new physical insights of this specific manuscript are very limited.”

The first point concerns the originality of our work and the connection to ref. 47. We would like to stress that there are many important differences between these two works.

  • Ref. 47 focuses on a magnetic field rotating in a plane perpendicular to the ribbon, which is fully equivalent, to the 'traditional' case of Rashba plus uniform magnetic field perpendicular to the ribbon, as well as on a field rotating in the plane of the ribbon, equivalent to an out-of-plane Rashba plus an uniform in-plane field. Our main results are for in-plane Rashba plus an in-plane field; this situation is not covered by Ref. 47 and our results are thus different and original. Moreover, in the new version of the paper we address also the case of the nanoribbon with in-plane Rashba and also with perpendicular field, and we find that the direction of the magnetic field is important and gives rise to important differences in the results

  • For the case of the Rashba/Zeeman, Ref 47 works in the limit of the single-band, also considers only a small portion of the phase diagram, that of the lowest mu and B. Using numerical methods and the Majorana polarisation we are able to recover the entire phase diagram, for all values and mu and B, and also to take into account the contribution of all the bands in the spectrum. We also study the quantitative dependence on the spin-orbit parameter which proves to be quite significant and non-intuitive.

  • Also for the Rashba/Zeeman analysis, in 47 the authors consider only zigzag ribbons, while in our work we also study the armchair ones, and we stress the differences between the two, which turns out to be quite important.

  • Ref. 47 needs an extra KK' coupling for the particular case of zigzag gapless nanoribbons, in order to generate a gap in the spectrum to have access to Majorana states, while our approach predicts Majoranas even when such a coupling is absent, based solely on the clean graphene band structure. We have added a subsection and a figure in the paper to address the peculiarities of gapless ribbons when studying the formation of Majoranas as a function of the ribbon width.

  • last, we should note that the qualitative description of Majorana formation in Ref. 47 omits certain important details, and thus a more detailed and quantitative analysis of the formation of Majoranas in graphene strips is quite in order.

All these differences are now stressed more clearly in the present version of the manuscript.

“2.) Large parts consist of blackbox numerical analysis without clear physical pictures. For example, the physics behind observations such as the importance of the ribbon width for the stability of the Majoranas is hardly discussed.”

We agree that it would be interesting to study the effects of the width of the ribbon, and a new subsection and a figure have been added in which such effects are discussed in detail and explained from a more physical perspective.

“3.) No immediate experimental relevance. As a main motivation for the the present study, the authors state the need for finding experimentally more accessible/realistic material systems hosting Majorana bound states. However, as reflected in the conclusion "As a future extension of the present work we propose to perform fully-realistic studies...", the manuscript falls short of really answering relevant experimental questions such as localization and coherence properties of the Majoranas, realistic numbers for the involved energy gaps, etc.”

In what concerns the experimental feasibility of our model, we need to point out that most of the works in this field consider systems as unrealistic, or even more unrealistic than the one we present here. The system described in Ref. 47 is not any more feasible than ours. Thus we do not believe that this may be a valid argument against the publication of our paper. Here we actually make quite a few steps in considering some possible realisations of this setup in experiments, as well as take into account realistic values of the parameters, but of course our analysis cannot capture all the microscopic effects that may arise in a given setup. We believe that our analysis provides however an important and necessary step towards a realistic modelling of the graphene nanoribbons, as well as proves that the technique of tight-binding associated with that of the Majorana polarization are the appropriate tools towards the study of the formation of Majoranas in such systems, which cannot be done using methods such as the ones in Ref. 47, if we are interested in taking into account both the width of the ribbon, as well as the effects of the edge types, of all the bands, and of an extended parameter space, as well as different configurations of non-uniform magnetic fields. Also the main point of this paper is to study the possibility to form Majoranas, not their stability, coherence, energy gaps, etc, those are all relevant questions and are achievable via the techniques we present here, but beyond the scope of this present work; our paper is already quite heavy in terms of the amount of results, thus we believe that a subsequent study is in order to address all these issues.

The main point is that our technique may give one access to a fully experimentally relevant situation even if our limited numerical abilities prohibit us from pursuing it in detail in the present work. Here we only provide an example of the full results that can be obtained using such techniques, as well as extract some results that hold generically for graphene strips.

We have modified our conclusions accordingly.

“I ask the authors to clearly state how the phase diagram in Fig. 2b) is computed.”

We have added a short description of the methods used to obtain the phase diagram in Fig. 2b.

“Furthermore, a clear physical discussion of how the finite size of the ribbons affects their topological properties would be in order, given that this point is strongly emphasized in the introduction.”

As described above, this is addressed in the extra subsection added to the present version of the paper.

We have thus made clearer the originality of the paper and added new discussions concerning the effect of the ribbon width, as well as the direction of the magnetic field, thus we believe that our paper can now be published in SciPost.

Yours sincerely, The authors Cristina Bena and Vardan Kaladzhyan

---

## Round 3 · List of Changes

1. All these differences with Ref. 47 are now stressed more clearly in the present version of the manuscript.
  2. We have added a short description of the methods used to obtain the phase diagram in Fig. 2b, as requested by the referee.
  3. An extra subsection with describing dependence of our results on the width of the ribbons has been added.
  4. We have also added a small subsection emphasising the difference between in-plane and perpendicular magnetic fields.

---

## Editorial Decision

published